# Hypoalbuminemia as Surrogate and Culprit of Infections

**DOI:** 10.3390/ijms22094496

**Published:** 2021-04-26

**Authors:** Christian J. Wiedermann

**Affiliations:** 1Institute of General Practice, Claudiana–College of Health Professions, 39100 Bolzano, Italy; christian.wiedermann@am-mg.claudiana.bz.it; 2Department of Public Health, Medical Decision Making and HTA, University of Health Sciences, Medical Informatics and Technology, 6060 Hall in Tyrol, Austria

**Keywords:** albumins/pharmacology, hypoalbuminemia, infections, innate immunity, bioactive lipid mediators, community-acquired pneumonia, bacteremia, sepsis, healthcare-associated infection

## Abstract

Hypoalbuminemia is associated with the acquisition and severity of infectious diseases, and intact innate and adaptive immune responses depend on albumin. Albumin oxidation and breakdown affect interactions with bioactive lipid mediators that play important roles in antimicrobial defense and repair. There is bio-mechanistic plausibility for a causal link between hypoalbuminemia and increased risks of primary and secondary infections. Serum albumin levels have prognostic value for complications in viral, bacterial and fungal infections, and for infectious complications of non-infective chronic conditions. Hypoalbuminemia predicts the development of healthcare-associated infections, particularly with *Clostridium difficile*. In coronavirus disease 2019, hypoalbuminemia correlates with viral load and degree of acute lung injury and organ dysfunction. Non-oncotic properties of albumin affect the pharmacokinetics and pharmacodynamics of antimicrobials. Low serum albumin is associated with inadequate antimicrobial treatment. Infusion of human albumin solution (HAS) supplements endogenous albumin in patients with cirrhosis of the liver and effectively supported antimicrobial therapy in randomized controlled trials (RCTs). Evidence of the beneficial effects of HAS on infections in hypoalbuminemic patients without cirrhosis is largely observational. Prospective RCTs are underway and, if hypotheses are confirmed, could lead to changes in clinical practice for the management of hypoalbuminemic patients with infections or at risk of infectious complications.

## 1. Introduction

Albumin is abundant in human blood, and research has identified a wide range of putative roles for the protein in modifying inflammation, maintaining vascular endothelial integrity and acid-base balance, and ligating endogenous and exogenous compounds [1]. Albumin can offer protection from inflammatory processes and the associated damage to microcirculation and tissues [2]. Albumin is involved as a carrier molecule in the distribution and elimination of drugs [3]. The kinetics of albumin involve trans-capillary leak and breakdown, leading to hypoalbuminemia, which is associated with worse outcomes in a broad spectrum of conditions [4]. Hypoalbuminemia is often due to inflammation but can also be caused by hepatocyte damage and decreased albumin synthesis, dietary insufficiency of amino acids, or increased excretion of albumin [4].

The therapeutic use of human albumin solution (HAS) in critically ill or surgical patients and to correct hypoalbuminemia has been studied for decades. HAS was introduced as a biopharmaceutical drug for plasma expansion in the 1940s but randomized controlled trials (RCTs) at low risk of bias to investigate its effects first started in the 1970s. Conflicting results from predominantly small studies of fluid therapy and volume resuscitation, together with increasing competition from synthetic colloids, contributed to great variability in the use of HAS in different countries and healthcare settings [5]. Before the turn of the century, clinical use of colloids shifted from albumin to artificial colloids, in particular hydroxyethyl starch solutions. This, combined with consideration of the relative safety and costs of colloids in general when compared with crystalloids tended to decrease the overall use of HAS [6]. More recently, however, growing evidence of safety and efficacy has led to increasing interest in the therapeutic potential of HAS [7].

The role of therapy with HAS in patients with liver disease is well established. Guidelines and consensus recommendations favor use of HAS over either crystalloids alone or synthetic colloids (in particular hydroxyethyl starch) in spontaneous bacterial peritonitis (SBP) and other complications of cirrhosis [8,9]. In clinical studies, the use of HAS in cirrhotic patients with peritoneal infection led to significant reductions in mortality and renal impairment [10]. Recent evidence suggests that HAS administration in combination with antibiotics in patients with cirrhosis and infections other than SBP reduced occurrence of new bacterial infection during follow up compared with antibiotics alone [11]. The mechanisms involved in this effect are not clear and final data from further studies investigating the effects of HAS in preventing infections in patients with advanced liver disease are not yet fully available [12,13].

This review focusses on albumin in the context of infections, including the roles of hypoalbuminemia and of adjunctive treatment with HAS. A non-systematic search of the literature was performed in EMBASE with the terms “(‘hypoalbuminemia’/exp OR hypoalbuminemia) AND (‘infection’/exp OR infection)” to identify studies published since 2016 until October 2020 and by manually searching reference lists for literature published before 2016. Approximately 2600 search hits were screened across a range of different disease settings, and identified studies were included in this report when deemed relevant.

## 2. Etiology of Hypoalbuminemia

Albumin represents almost three-quarters of the antioxidant capacity of serum [14]. Hepatic synthesis maintains circulating albumin, which can access the interstitium and tissues where it acts as an antioxidant and free radical scavenger [4]. Eventually, the liver reduces or resynthesizes albumin to restore its antioxidant activity and there can be an upregulation in the synthesis and anti-inflammatory functions of albumin depending on the severity of inflammation (Figure 1) [4]. However, breakdown of albumin is also increased in inflammatory states, which can lead to hypoalbuminemia. despite increased albumin synthesis [4]. A combination of the inflammatory response and increased capillary escape of albumin may also limit its oxidative influences in the interstitial space in infection, serious illness, and trauma, but at the same time provide substrate for tissue regeneration [15].

### 2.1. Albumin Kinetics

Albumin synthesis in the liver is partly controlled by the oncotic pressure in the hepatic interstitium. On the one hand, serum albumin mass is influenced by the hepatic fractional synthesis rate, which may be reduced in inflammatory conditions but can also be normal or even increased. On the other hand, capillary escape [16], increased breakdown [17], and external losses occurring in nephrotic syndrome, protein-losing enteropathy due to lymphatic blockage, and mucosal disease or extensive burns, reduce the half-life of serum albumin [4]. In liver failure, fractional synthesis rate is mainly decreased because production of albumin is affected by liver function [18]. Low serum albumin is a parameter of the Child-Pugh Scoring system for assessing disease severity in decompensated cirrhosis, and an indicator of prognosis [19].

### 2.2. Hypoalbuminemia in Acute Phase and Chronic Inflammation

Even in otherwise healthy individuals, chronic inflammation is a driving force of reduced circulating albumin levels. Hypoalbuminemia reflects the physiological stress from disease-related inflammation, may develop quickly even in adequately nourished patients after trauma or acute illness, and commonly occurs in chronic inflammatory conditions due to increased vascular permeability and interstitial volume [4]. With advancing age, a progressive, systemic, but subclinical, inflammatory process develops which has been named ‘inflammaging’ [20]. This low-grade inflammatory state potentially contributes to chronically reduced serum albumin in patients with no clinically apparent inflammatory disease. In the elderly, increased concentrations of IL-6 and TNFα, are associated with hypoalbuminemia, which in turn is a predictor of mortality [21].

The severity of hypoalbuminemia reflects the severity of inflammatory stress in both acute and chronic states. Chronic conditions associated with hypoalbuminemia also complicate hospital treatment by mechanisms described as ‘second hit’ phenomena [22]. In established chronic diseases, inflammation contributes to albumin loss, which can lead to poorer outcomes [23]. A systemic inflammatory response and low serum albumin level is associated with negative prognosis in patients with cancer, most likely through suppression of the immune response [24]. Although the strength of the correlation is not clear, severe inflammation is generally associated with progressively lower serum albumin levels and, most importantly, low levels of serum albumin are associated with worse recovery in acute conditions involving systemic inflammatory responses [25]. It may therefore be that preexisting hypoalbuminemia contributes to the risk of acute infectious diseases, and that acute loss of albumin in systemic inflammatory reactions further complicates the clinical course of all trauma, medical, and surgical conditions.

## 3. Bio-Mechanistic Relationship between Hypoalbuminemia and Infection

Infections are the most common trigger of acute phase hypoalbuminemia. Distinguishing between appropriate development of hypoalbuminemia for controlling a widespread infection and excessive capillary leakage is challenging, so treatment is often focused on addressing the underlying cause without replacing albumin. The effects of albumin in infectious disease are predominantly indirect and hypoalbuminemia may reflect the poor health of the patient. Equally, critically ill patients are likely to have some immune-mediated defects that are not secondary to hypoalbuminemia, and overall, it seems that hypoalbuminemia is a reflection of disease or trauma-related inflammation [4]. There is bio-mechanistic plausibility for a link between hypoalbuminemia and increased risks of primary and secondary infections, although direct proof is not available to establish causal relationships. Most of the in vitro evidence of a direct immunomodulatory effect of albumin is old (published well before 2016) and outside of the scope of the literature search used to prepare this review. Nevertheless, albumin physiology plays a critical role in both an effective host immune response to pathogens and the devastating effect of immune dysregulation characterized by cytokine storm [26]. The combination of acute-phase hypoalbuminemia and endothelial cell death can lead to capillary leak syndrome, with renal dysfunction further contributing to unfavorable edema formation—changes that are similar to those observed in patients with cancer who are treated with high-dose interleukin (IL)-2 [27].

Some studies suggest that therapeutic use of HAS may beneficially affect the pathophysiology of immune dysregulation-initiated inflammatory events. Interestingly, in a prospective randomized trial undertaken to compare crystalloid and colloid fluid resuscitation for patients receiving bolus IL-2-based therapy for metastatic cancer, 5% HAS alleviated hypoalbuminemia despite vascular leak and the incidence of oliguria was markedly reduced by albumin infusion (2.8% vs. 32.5% with normal saline) [28]. Similarly, in a multi-center RCT involving 193 patients with severe liver disease and sepsis unrelated to spontaneous bacterial peritonitis, patients were randomized to receive either antibiotics plus HAS (1.5 g/kg on day 1 and 1 g/kg on day 3), or antibiotics alone. HAS did not improve renal function or survival at 3 months; however, the short-term use of HAS significantly delayed onset of renal failure [29].

### 3.1. Albumin and Immune Function

The ability of the albumin molecule to bind a variety of ligands is central to its immunomodulatory effects [1]. Albumin presents a high degree of microheterogeneity, which is affected by physiological and pathological processes [30]. The oncotic effect of serum albumin derives from its molecular weight and negative charge, but non-oncotic properties, including protection against oxidative damage, and binding activities for mediators, drugs, and toxins are related to other aspects of its molecular structure. The amino acid residue at position 34 from the N-terminus is a cysteine with a mercapto group (sulfhydryl group) (Figure 2), which deoxidizes other substances according to the degree of oxidative stress. Oxidative damage subsequently impairs the binding properties of albumin as found in healthy [31,32] and diseased subjects, including patients with cirrhosis of the liver, kidney disease, and sepsis [33].

In patients with decompensated cirrhosis, serum albumin shows structural abnormalities, most often involving reversible oxidation and glycation [35], which lead to impaired binding capacity in albumin and increased risk of liver failure [36]. By combining liquid chromatography-electrospray ionization-mass spectrometry and standard methods of albumin determination, ‘effective’ albumin has been defined as the albumin isoform presenting structural and functional integrity as differentiated from ‘total’ albumin, which includes modified albumin. Most importantly, it has been suggested that serum levels of ‘effective’ albumin may be a better prognostic indicator than ‘total’ albumin in patients with decompensated cirrhosis [36].

Immune responses to invasive pathogens include elevations of nonspecific markers of inflammation such as C-reactive protein (CRP) [37]. A variety of mediators provide regulatory signals that coordinate antimicrobial effector cells of the innate and adaptive immune functions [26]. Trans-capillary leakage of serum albumin correlates with infection severity and increases interstitial levels of albumin [16]. The antioxidative and other non-oncotic properties of albumin may be of critical importance in antimicrobial defense and injury repair mechanisms, and more significant hypoalbuminemia can therefore have systemic effects and cause collateral damage to vital organ systems [1,3,38]. Experimental evidence suggests that antioxidant treatment after injury may be beneficial against bacterial infection in suppressing second-hit responses [39].

#### 3.1.1. Effects of Albumin on Leukocytes

Neutrophilic granulocytes form extracellular webs of chromatin, microbicidal proteins and oxidant enzymes known as neutrophil extracellular traps (NETs) as an immune defense mechanism to contain infections and albumin inhibits NETosis by the scavenging of activators such as bacterial lipopolysaccharides [40]. Infectious diseases can be characterized by increased serum levels of cell-free DNA, reflecting NETosis and decreased levels of albumin [41,42]; however, the degree of inverse correlation still needs to be investigated. Casulleras et al. [43] used peripheral blood leukocytes from patients with cirrhosis of the liver and showed that albumin administration inhibited cytokine production induced by bacterial 5’-C-phosphate-G-3′-DNA. Albumin was taken up by leukocytes and localized in endosomes, where it inhibited Toll-like receptor signaling. In patients with acutely decompensated cirrhosis, albumin administration has been shown to reduce inflammation, and may therefore involve a direct effect of albumin on leukocytes [44].

#### 3.1.2. Interactions of Albumin with Sphingosine 1-Phosphate and Heparin-Binding Protein

Preclinical studies demonstrate that albumin is more effective than hydroxyethyl starch and other artificial colloids in preserving the endothelial glycocalyx, reducing vascular permeability, and reducing adhesion of platelets and leukocytes [45]. Low protein content in plasma causes matrix metalloproteinase (MMP) cleavage of the glycocalyx components from the underlying endothelium [45].

Protection against endothelial glycocalyx shedding might be mediated by the protein-bound lipid mediator sphingosine 1-phosphate (S1P) through inhibition of MMP cleavage of the endothelial glycocalyx and restoration through mobilization of intracellular glycocalyx components [45]. Circulating S1P has been found to be a key regulator of lymphocyte trafficking, endothelial barrier function, and vascular tone [46] Erythrocytes and platelets are major sources of S1P in the body, and albumin can facilitate the release of and transport S1P [46]; in the absence of albumin, S1P release from erythrocytes is dramatically reduced and it seems that albumin plays an import role in providing a significant amount of S1P to the endothelium. [47].

Heparin-binding protein (HBP) is also involved in the pathophysiology of severe bacterial infections and may have a role in their management [48]. A high level of HBP in plasma might be a useful diagnostic marker for detecting organ dysfunction among patients ranging from suspected infections to septic shock [49]. In clinical studies, high plasma HBP levels in patients with sepsis were associated with a higher early fluid balance, increased organ failure within 5 days, and higher 28-day mortality [50]. Fisher et al. [51] reported that patients in septic shock in whom acute kidney injury developed or worsened had significantly higher HBP-to-albumin ratio and HBP levels in plasma than patients with no acute kidney injury. The highest HBP-to-albumin ratio and HBP levels were significantly associated with development or worsening of acute kidney injury, positive fluid balance, and levels of inflammatory cytokines. In vitro albumin inhibited HBP-increased endothelial cell permeability, suggesting that HBP may be a mechanism of efficacy in septic shock [51].

Interestingly, in coronary bypass surgery, incidence of acute kidney injury is elevated and plasma levels of HBP are known to increase, [52,53] preoperative serum albumin levels are inversely related to postoperative risk of acute kidney injury development, [54] and in an RCT, preoperative correction of hypoalbuminemia with administration of HAS significantly reduced the occurrence of acute kidney injury [55].

#### 3.1.3. Interaction of Albumin with Bioactive Lipid Mediators

The cell membrane is rich in polyunsaturated fatty acids that give rise to both pro-inflammatory and anti-inflammatory eicosanoids [56]. Hypoalbuminemia might alter the metabolism of arachidonic acid, and lower arachidonic acid level has been reported to be an important determinant of outcome in sepsis patients [57]. Albumin enhances the formation of anti-inflammatory lipoxins, resolvins and protectins that facilitate resolution of disease processes (Figure 3) [58]. This is consistent with the findings of previous studies which reported that elevated prostaglandin E_2_ (PGE_2_) combined with hypoalbuminemia mediates immunosuppression in patients with acute decompensated and end-stage liver disease, which can be reversed with 20% HAS infusions [12].

Acute inflammatory conditions have been associated with multisystem organ failure in obese patients, and animal studies showed that release of polyunsaturated fatty acids is pro-inflammatory [59]. Interventions that prevented an increase in serum unsaturated fatty acids in mice prevented renal injury, lung injury, systemic inflammation, hypocalcemia, reduced pancreatic necrosis, and mortality [60]. Albumin can bind unsaturated fatty acids and reduce inflammation injury [61]. The observed regulation of lipotoxicity by albumin may be particularly relevant in coronavirus disease 2019 (COVID-19) patients with obesity and significant hypoalbuminemia [62].

## 4. Hypoalbuminemia and Disease Outcomes

Inflammation can affect the success of treatment because it reduces the effectiveness of biological responses in trauma or disease. In many chronic, progressive conditions, hypoalbuminemia worsens along with the severity of inflammation and risk of mortality [63]. Preexisting hypoalbuminemia is a prognostic indicator of worse outcome for a broad spectrum of diseases ranging from medical conditions [64,65,66] to surgery [67]. Evidence in support of a prognostic role of serum albumin level is abundant and established for orthopedic [68,69,70,71], cardiovascular [72,73], gynecologic [74,75], and visceral surgery [76,77,78].

Serum levels of albumin predict outcome in first-hit acute inflammatory conditions such as primary trauma [79], burns [80], or acute infections [81]. Acute inflammation elicits an acute phase reaction characterized by changes of albumin and other markers of inflammation [82] including CRP, which increases within hours after major surgery [83]. Circulating CRP levels predict outcome in surgery and acute medical conditions [84].

There is a close correlation between increases in CRP and decreases in serum albumin levels. In patients undergoing elective open colorectal surgery, it has been reported that preoperative CRP was useful in predicting the development of hypoalbuminemia on postoperative days 3 and 7 [85]. Serum albumin concentration on the first postoperative day has also been shown to be a better predictor of surgical outcome than other preoperative risk factors [86]. The predictive value of serum albumin levels is the rationale to include serum albumin levels in physiological scoring systems for critically ill hospitalized adults such as APACHE III [87] and also in a simplified critical illness severity scoring system (CISSS) that was validated recently [88].

To increase the predictive value of albumin and CRP, the two markers have been combined, with the ratio of serum levels of albumin and CRP known as the ‘Glasgow Prognostic Score’ [89]. If and how the Glasgow Prognostic Score will be able to increase the predictive power for outcome is being studied intensively in different fields and patient populations [90]. Detailed discussion of this and other biomarker ratios is beyond the scope of this review.

Data from the first US Health and Nutrition Examination Survey (NHANES I) show that albumin concentrations are related to lean body mass in healthy males [91]. Older people with low albumin levels can show loss of muscle mass, and hypoalbuminemia is sometimes considered to be an indicator of malnutrition, but there are other factors that affect serum albumin levels [21]. A study in life insurance applicants showed that across a wide range of values, albumin levels independently predicted mortality risks in healthy men and women of all ages [63], and other data show that, regardless of its cause, low serum albumin is a strong predictive factor for mortality and morbidity [92].

Liver cirrhosis, malnutrition, nephrotic syndrome, and sepsis are known to be strongly associated with hypoalbuminemia and frequently require hospitalization [92], and there are links between hypoalbuminemia on admission and in-hospital mortality in a range of illnesses, including sepsis [93]. Because low serum albumin levels reflect severity of inflammation, in seriously ill patients, changes in serum albumin levels in the absence of albumin administration can indicate either recovery or deterioration [4].

## 5. Hypoalbuminemia and Prognosis in Infections

Associations of serum albumin levels with development and severity of infectious diseases may be sufficiently explained by the effects of systemic inflammation on albumin kinetics making low serum albumin levels a surrogate marker. If low serum albumin levels can be shown to directly affect innate immunity and antimicrobial defense, associations with infectious diseases might also lead to the identification of low albumin mass as culprit, causally contributing to both acquisition and development of complications of infections [94].

The prognostic value of low serum albumin levels has been demonstrated not only for the development and complications of acute viral, bacterial, and fungal infections [95], but also for infectious complications in chronic conditions including malignant diseases [66,96], chronic inflammatory diseases [97,98], diabetes [95], hemodialysis [99], and solid organ transplantation [100]. In chronic infectious diseases, hypoalbuminemia is associated with disease reactivation/recurrence and secondary infections [101,102,103].

As shown in patients during the initial phase of bacteremia, after an initial increase in CRP, levels drop over several days, whereas after the sudden drop of serum albumin levels due to trans-capillary leakage, serum levels remain low for a much longer period [16]. Patients who present with community-acquired bacterial or viral infections are more likely to require intensive care unit (ICU) treatment if serum albumin levels on admission to hospital are low [104,105]. Patients with colorectal cancer who develop infectious complications after laparoscopic surgery have more pronounced postoperative hypoalbuminemia on days 2 and 3 than patients with uncomplicated surgery [106], suggesting that measurement of serum albumin levels may be helpful in the early recognition of postoperative infection. Hypoalbuminemia is associated with the severity of different kinds of infections, including those caused by *Pseudomonas aeruginosa*, *Clostridium difficile*, and influenza [101,107,108], and albumin levels are predictive of infection complications as shown in scrub typhus [109].

In particular, hypoalbuminemia is an independent risk factor for mortality in *Clostridium difficile* infection, along with older age, immunosuppression, and antibiotic use [107,110,111,112,113]. Bell et al. reported that hypoalbuminemia was a risk factor for increased risk 90-day incidence of *Clostridium difficile* infection after spine surgery, along with preoperative fluoroquinolone use, advanced age, chronic kidney disease, and decompensated chronic liver disease [107]. Knafl et al. reported on 144 consecutive symptomatic patients with a positive stool test for *Clostridium difficile* in whom ATLAS score (Age, Temperature, Leukocyte count, Albumin, Systemic antibiotics, Serum creatinine) was calculated [112]. This found that both serum albumin and ATLAS were predictors of disease complications and mortality from *Clostridium difficile* infection while only serum albumin was significantly associated with 90-day disease recurrence.

It is known that the effects of *Clostridium difficile* are related to production in the intestine of protein toxins known as TcdA and TcdB [114] and it has been shown that human serum albumin binds to these toxins and causes their proteolytic cleavage outside intestinal epithelial cells [115]. This occurs at physiological concentrations and impacts the ability of the toxins to enter the host cells and affect GTPases to produce cytotoxic effects, thereby providing a possible explanation for the observed correlation between serum albumin levels and the severity of *Clostridium difficile* infection [115].

### 5.1. Sepsis

In sepsis, acute and chronic states of low serum albumin are independently associated with increased risk of mortality. Sepsis is a common cause of mortality in ICUs, and low serum albumin levels in the acute phase are associated with increased risks of severity [116] and death in patients who develop severe sepsis and organ failure [81,117,118,119]. The predictive value of low serum albumin is independent of the site of infection [120,121,122] and independent of the patients’ age, even considering newborn and pediatric patients with sepsis [123,124,125]. Serum albumin of 2.45 g/dL was identified as the cut-off value to define hypoalbuminemia that was optimal for the prediction of short and long-term mortality in patients with septic shock [126].

Preexisting chronic hypoalbuminemia is an established component in the ‘predisposition, infection/insult, response, and organ dysfunction’ (PIRO) sepsis staging system (Table 1) [127]. In a Danish population-based study in 1844 patients with community-acquired bacteremia, a single plasma albumin measurement on the date bacteremia was identified was a better predictor of short-term mortality than a sepsis severity score [38]. For each 0.1 g/dL decrease in serum albumin level, the odds ratio (95% confidence intervals) of mortality in the period of 0–30 days after bacteremia was 0.86 (0.84–0.88). In that study, the latest albumin measurement was taken 8–30 days before the date of bacteremia in 422 patients, and a higher proportion of patients died within 30 days if the albumin level decreased between days 8–30 prior to bacteremia and the eventual date bacteremia was identified [38].

**Table 1 ijms-22-04496-t001:** Score value of the ‘predisposition, infection/insult, response, and organ dysfunction’ (PIRO) staging system for community-acquired sepsis in the emergency department. Adapted from Chen and Li 2013 [127].

Component	Parameter	Threshold	Weighted Score
Predisposition (P)	Age	>70	2
	Chronic obstructive pulmonary disease	Yes	1
	Serum albumin	<2.5 g/dL	2
Infection (I)	Central nervous system infection	Yes	9
Response (R)	Temperature	<36 °C or >38.5 °C	2
	Procalcitonin	>0.5 ng/mL	7
	Brain natriuretic peptide	>113 pg/mL	7
Organ dysfunction (O)	Troponin I	>0.075 ng/mL	2
	Mean arterial pressure	<70 mmHg	2
	Glasgow coma scale	≤14	4

In inflammatory bowel disease with chronic inflammation, surgical stress increases the risk of postoperative sepsis if preoperative serum albumin levels are low [128,129]. Similarly, in hemodialysis, hypoalbuminemia affects the risk of sepsis in catheter-related bloodstream infection [130]. The mortality risk increases in septic patients if other conditions/markers that have predictive value are combined with serum albumin levels, as shown for ratios between serum albumin and blood glucose [131], serum albumin and lactate levels [132,133], or serum albumin and immature neutrophil counts [134].

### 5.2. Community-Acquired Pneumonia and Bacteremia

Hypoalbuminemia is associated with acquisition and severity of community-acquired pneumonia (CAP) [135,136,137]. In a total of 3463 patients hospitalized with CAP, the median serum level of albumin on admission was 3.1 g/dL (interquartile range 2.8 to 3.5), with decreased levels significantly associated with increased time to clinical stability, longer hospital stay, admission to the ICU, mechanical ventilation and 30-day mortality [135]. Serum albumin levels are one of the criteria in the SMART-COP (systolic blood pressure, multi-lobar chest radiography involvement, albumin level, respiratory rate, tachycardia, confusion, oxygenation, and arterial pH) prediction rule that is validated for identifying patients with CAP who need vasopressor and/or respiratory support [138].

The development of bacteremia in patients with CAP is associated with increased risk of septic shock or mortality [139], and hypoalbuminemia is an independent risk factor for the development of bacteremia in patients with CAP [140] and bacteremia-related mortality [141,142,143].

### 5.3. Healthcare-Associated Infections

Healthcare-associated infections have been defined as those occurring while receiving (or within 30 days of receiving) healthcare, or in a hospital or healthcare facility that first appear ≥48 h after admission [144]. Many healthcare-associated infections affect surgical sites, but others are due to central line bloodstream infections, catheter-associated urinary tract infections, ventilator-associated pneumonia, septicemia, and gastrointestinal infections. [144]. *S. aureus*, *Enterococcus* spp., *E. coli* and coagulase-negative *Staphylococci* are among frequently detected microorganisms and up to 20% of microorganisms are multidrug resistant [144].

In the elderly, hypoalbuminemia is among the clinical parameters associated with poor clinical outcome of both community-acquired and healthcare-associated bacteremia [16,145,146,147]. Similar findings on the predictive value of hypoalbuminemia are reported for patients with bacteremia due to acute bacterial infections of the skin [148]. Low serum levels of albumin along with inappropriate antibiotic therapy are associated with increased 30-day mortality in patients with vancomycin-resistant Enterococcus bacteremia [149]. As described earlier, hypoalbuminemia is an independent risk factor for mortality in *Clostridium difficile* infection, along with old age, immunosuppression, preexisting conditions including bloodstream and preoperative infection, as well as antibiotic use [107,110,111,112,113].

#### 5.3.1. Surgical Site Infection

Surgical site infection has been defined as infection related to a surgical procedure that occurs at or near the surgical incision within either 30 days or within 90 days if prosthetic material is implanted [150,151]. Antimicrobial prophylaxis aims to prevent surgical site infections and should be combined with infection-control strategies, and perioperative management of the underlying medical condition of the patient (which may involve reduced serum levels of albumin) may also have an impact on surgical infection rates [152]. However, it has not yet been reported whether hypoalbuminemia impairs the effectiveness of antimicrobial prophylaxis.

An increased risk of surgical site infection is associated with many patient-related factors that have been associated with hypoalbuminemia, including age, nutrition, obesity, diabetes mellitus, smoking, existing infections, immunosuppressant or corticosteroid therapy, undergoing a recent surgical procedure, duration of hospitalization before surgery, and colonization with drug-resistant bacteria [150]. Hypoalbuminemia can be present before surgery due to illness, insufficient hepatic synthesis, or poor preoperative nutritional status, and can occur after surgery due to inflammation, bleeding or capillary leakage [153].

Surgical site infection is a rare complication in hernia repair surgery, with an incidence of 0.4% (254/57,951) in a large database study; serum albumin was not independent of diabetes, body mass index >35, and smoking as a predictor of the risk of this complication [154].

An albumin level <3.5 g/dL was associated with ~2.5-fold increased risk of surgical site infections in orthopedics [155]. Hypoalbuminemia was independently associated with surgical site infections in several observational studies involving orthopedic surgery [79,155,156], but this has not yet been confirmed in prospective cohort studies.

Surgical site infection in spine surgery is associated with both patient- and procedure-related risk factors. Patient-associated risk factors, many associated with hypoalbuminemia, include diabetes mellitus, obesity, subcutaneous fat thickness, comorbidities, tobacco use, and malnutrition [157]. Hypoalbuminemia is more common in revision spine surgery for septic reasons than in aseptic revision surgery [158] and increased the length of hospital stay for postoperative acute surgical site infection after spinal surgery [159].

In gynecologic surgery, a registry study in 777 women undergoing surgery for vulvar cancer reported that major postoperative wound complications occurred with an incidence of 10.4% and were associated with preoperative hypoalbuminemia [160]. In another registry study of gynecologic cancer surgery, 369/6854 (5.4%) of patients were identified with surgical site infections which were associated with longer stays in hospital, more frequent need for reoperation, and higher rates of sepsis and wound dehiscence [161]. Hypoalbuminemia was among the predictors for occurrence of both deep and organ-space surgical site infections.

In head and neck surgery, postoperative wound infections after laryngectomy cause substantial morbidity. In a registry study of more than 2000 patients who underwent laryngectomy, the overall wound complication rate was 10.0% and in adjusted analyses hypoalbuminemia (odds ratio, 1.90; 95% confidence interval, 1.32–2.74) was independently associated with postoperative wound complications; other risk factors were an operative time of more than 10 h, prior exposure to radiation therapy, presence of diabetes, preoperative anemia, and thrombocytosis [162]. In a study of cancer patients undergoing head and neck surgery, there was a surgical site infection rate of 104/399 (28%); patients with preoperative hypoalbuminemia (<3.3 g/dL) had a 3-fold higher risk of surgical site infection [163]. In oral cancer surgery, it has also been reported that early postoperative hypoalbuminemia (<2.5 g/dL) is an independent risk factor for the development of surgical site infections. The mean time to infection manifestation was 10 days and the duration of hospital stay was negatively correlated with postoperative albumin, suggesting that identification of hypoalbuminemia could be used to alert clinicians about postoperative care needs [164].

In vascular surgery, it has been shown that surgical site infections are frequent after lower extremity bypass and are associated with early graft failure and sepsis [165]. In a registry study of 7595 bypasses, surgical site infections occurred in 11% of overall cases and preoperative hypoalbuminemia was an independent risk factor [165]. Furthermore, in 106 consecutive patients with peripheral arterial disease who underwent lower limb amputations, the 30-day mortality rate was 7.6%, with multivariate analysis showing that a low serum albumin concentration was a risk factor (hazard ratio, 3.87; 95% confidence interval, 1.12–16.3) [166]. The most frequent causes of 30-day mortality were pneumonia, sepsis, and cardiac death.

In patients with intra-abdominal infections, host-related factors dominate over the type, extent, and source of infection in determining prognosis, and low serum albumin is one of few identified risk factors for mortality [167]. Preoperative or perioperative hypoalbuminemia is an independent risk factor for the development of surgical site infection in patients undergoing gastrointestinal surgery [168,169,170,171,172,173]. Inclusion of preoperative serum albumin levels enabled development of a validated stratification scoring system to predict accurately the risk of surgical site infection after esophagectomy in patients with esophagus carcinoma [173]. Patients with hypoalbuminemia are at increased risk of developing surgical site infections after gastric [170] and colorectal [172] cancer surgery. In a retrospective study of 524 patients undergoing gastrointestinal surgery, preoperative hypoalbuminemia (<3 g/dL) was associated with an increased rate of deep surgical site infections compared with superficial infections [169]. Hypoalbuminemia was not identified as an independent risk factor for either incisional or organ-space infectious complications after liver resection surgery [174].

#### 5.3.2. Catheter-Related Bloodstream Infection

Hypoalbuminemia and anemia have been associated with poor prognosis in patients with bloodstream infection, which occur more readily in immunocompromised patients and those with a high APACHE Ⅱ score on admission [101]. Immunosuppression, anemia, and hypoalbuminemia are prominent risk factors of access-related bloodstream infection in patients with end-stage renal disease [99,130]. The combination of preoperatively increased CRP and decreased serum albumin levels (i.e., Glasgow Prognostic Score) was strongly associated with perioperative and postoperative central venous catheter-related bloodstream infection in colorectal cancer patients receiving intravenous parenteral nutrition [175]. Similarly, hypoalbuminemia at the time of port placement was a predictor of early port infections in adult patients with hematologic malignancies [176].

#### 5.3.3. Hospital-Acquired, Ventilator-Associated, and Healthcare-Associated Pneumonia

Hospital-acquired and ventilator-associated respiratory infections are the most prevalent nosocomial infections in ICUs [177]. Although hypoalbuminemia is not identified as an independent risk factor of nosocomial pneumonia, many patient-related risk factors are characterized by low serum albumin levels associated with acute or chronic severe disease and comorbidities [178]. Prolonged and/or inappropriate antibiotic treatment is among the most recognized procedure-related risk factors, and inadequate empirical antimicrobial therapy increases nosocomial pneumonia-related hospital mortality [179]. Hypoalbuminemia is associated with inappropriate antimicrobial treatment [180,181] and may be implicated by contributing to patient-related risk factors and other mechanisms relating to the pharmacokinetic and pharmacodynamic roles of albumin.

### 5.4. Coronavirus Disease 2019

Reduced serum levels of albumin are associated with poorer outcomes in patients with COVID-19 [41]. It has been reported that patients had different probabilities of disease progression based on age, residence in nursing home, comorbidities, obesity, respiratory rate and presence of respiratory symptoms, fever, lymphocyte count, troponin level, CRP level, and hypoalbuminemia [182]. The viral load of SARS-CoV-2 detected from patients’ respiratory tracts in a case series of 12 patients has been linked to lung disease severity, and serum albumin and markers of systemic inflammation were highly correlated to acute lung injury [183]. Hypoalbuminemia has also been reported to be present in pediatric patients with Kawasaki-like syndrome associated with COVID-19 [184,185].

## 6. Hypoalbuminemia and Antimicrobial Treatment

In a prospective cohort study evaluating the relationship between antimicrobial treatment for bloodstream infections and clinical outcomes among 492 patients requiring ICU admission, 174 patients received inadequate antimicrobial treatment and had a significantly increased hospital mortality rate. Multiple logistic regression analysis demonstrated that decreasing serum albumin concentration (1 g/dL decrements) was independently associated with the administration of inadequate antimicrobial treatment (adjusted odds ratio, 1.37; 95% confidence interval, 1.21 to 1.56; *p* = 0.014) [181]. Similarly, in 222 adult patients hospitalized with *Staphylococcus aureus* bacteremia, treatment failure at 30 days was independently associated with serum albumin levels (adjusted odds ratio, 0.88; 95% confidence interval, 0.80 to 0.96; *p* = 0.003) [180]. A recent prospective cohort study of clinical pharmacist-led antimicrobial treatment consultation in 2663 patients confirmed that hypoalbuminemia is independently associated with a decrease in effective response rate to antimicrobial treatment (adjusted odds ratio, 0.69; 95% confidence interval, 0.52 to 0.92; *p* = 0.012) [186].

Failure of antimicrobial treatment may occur because of failure to achieve adequate drug concentrations at the infection site because disease pathology alters pharmacokinetics. Protein binding is an important factor impacting pharmacokinetic parameters of antimicrobial drugs [187] because the unbound concentration is responsible for drug efficacy and potential drug toxicity [188]. Some antibiotics, such as β-lactams, aminoglycosides, and glycopeptides, have a higher risk of fluctuations in plasma levels because of their hydrophilic properties than more lipophilic antimicrobials such as macrolides, fluoroquinolones, tetracyclines, chloramphenicol, and rifampicin [189].

### 6.1. Antibiotics

#### 6.1.1. Beta-Lactams

Ceftriaxone is a ß-lactam antibiotic with extensive protein binding of 85% to 95% [190,191]. Low albumin levels increase the ceftriaxone unbound fraction, affect pharmacodynamic target attainment [192], and may lead to adverse drug reactions [193]. Hypoalbuminemia is a risk factor for treatment failure with cefotiam in bacteremia caused by gram-negative bacteria [194]. It has also been recommended that because of the high prevalence of hypoalbuminemia in critically ill patients, dose adjustments of flucloxacillin should be based on monitoring unbound concentrations to account for non-linear protein binding [195].

Ertapenem is highly protein-bound (92–95%), and data suggest that in critically ill patients with hypoalbuminemia, ertapenem dosed according to current guidelines is associated with higher mortality than alternative carbapenems [196]. Dose adjustment for ertapenem should therefore be guided by serum albumin concentrations, with dose intervals or continuous infusion considered with the aim of achieving optimal free drug concentrations in patients with severe hypoalbuminemia [197].

#### 6.1.2. Aminoglycosides, Glycopeptides, and Polymyxins

Serum albumin concentration is an important predictor of amikacin nephrotoxicity [198]. In septic, critically ill patients with severe hypoalbuminemia, administration of vancomycin loading doses can lead to potentially toxic serum concentrations [199]. Thrombocytopenia is one of the most common adverse reactions developed in critically ill adult patients treated with teicoplanin, and concurrent severe hypoalbuminemia significantly increases the risk [200].

Decreased serum albumin is a significant risk factor for the nephrotoxicity of polymyxin E (colistin), but this drug has been used to treat multidrug resistant Gram-negative pathogens [201]. Severe hypoalbuminemia independently predicted acute kidney injury during colistin therapy in a large cohort of patients with bloodstream infection [202].

### 6.2. Therapeutic Drug Monitoring

As dosing strategies designed to optimize antimicrobial pharmacokinetics/pharmacodynamics have been associated with clinical cure and survival in critically ill patients, therapeutic drug monitoring incorporating measurement of serum albumin levels has been recommended [203,204]. French guidelines suggest measuring serum albumin levels (or at least plasma proteins) at least once at the onset of treatment with beta-lactam antibiotics and in combination with therapeutic drug monitoring for additional interpretation of the results [205].

## 7. Therapeutic Effects of Human Albumin Infusion on Infection

Bacterial infection is one of four distinct clinical events that are precipitants consistently related to acute decompensation of cirrhosis [206]. Prevention of development of ascites and hepatorenal syndrome during an episode of SBP in patients with decompensated cirrhosis of the liver by short- and long-term infusion of HAS is an approved systemic therapy [207,208]. In a study of HAS administration in patients with cirrhosis and infections other than SBP, patients receiving albumin were less likely than controls to develop new bacterial infections during the follow-up period (9.8% vs. 24.6%, respectively; *p* < 0.03) [11]. Reduced systemic inflammation by HAS therapy in patients with decompensated cirrhosis may be responsible for the beneficial effects on outcomes, including in patients with infections other than SBP [209]. According to recent meta-analysis, there is no evidence of significant mortality or renal dysfunction benefits of using HAS for cirrhotic patients with extra-peritoneal infections [210].

A retrospective study of 306 older patients (≥65 years) with serum albumin levels <3.5 g/dL suffering femoral neck fracture and undergoing hip replacement examined the effects of HAS infusion alone or in combination with nutritional supplements to correct hypoalbuminemia (to ≥3.5 g/dL). Lower rates of surgical site infection (5.5% compared with 13.0% [adjusted odds ratio 0.40, 95% confidence interval, 0.17 to 0.91, *p* < 0.05]), and peri-prosthetic joint infection (2.8% compared with 9.9% [adjusted odds ratio 0.26, 95% confidence interval, 0.08 to 0.79, *p* < 0.05]) were seen in patients receiving albumin infusion combined with nutritional supplements. Postoperative day 5 serum albumin levels were 3.3 ± 0.16 g/dL for HAS infusions alone vs. 3.5 ± 0.15 g/dL for infusions combined with nutritional supplements (*p* < 0.001) [211]. Retrospective observational evidence has also suggested that hypoalbuminemia after spinal surgery is not associated with surgical site infection, and that supplementing human serum albumin increased the rate of surgical site infection [212]. The main limitations of such contradictory studies are the retrospective observational design, missing variables, and low patient numbers, precluding meaningful conclusions.

In a prospective, randomized, open-label, pilot study, 50 shock patients requiring norepinephrine (74% septic and 26% non-septic shock) with serum albumin levels <2.0 g/dL were allocated to receive either continuous 4% HAS or intermittent 20% HAS [213]. Continuous 4% HAS was effective in reducing healthcare-related infections compared with 20% HAS intermittently (2 vs. 13 episodes; *p* = 0.002), although circulating concentrations of serum albumin reached similar levels. At the same time, availability of antimicrobial vasostatin-I was increased, possibly related to the importance of continuously having available antioxidative capacity through therapeutic albumin, thus maintaining the antimicrobial domain 17–40 of vasostatin-I [213]. Although limited by small sample size, these data provide support for pharmacologic interventions to restore antioxidative capacity using a continuous infusion of 4% HAS. This, however, requires confirmation in further studies.

### Outlook on Future Studies of Hypoalbuminemia and Infection

Future clinical studies at low risk of bias will improve the level of evidence on the effects of HAS infusion in patients with hypoalbuminemia and infections. These include a prospective, phase III RCT that will study the effect of HAS administration on outcomes in hypoalbuminemic patients hospitalized with CAP [214]. The Albumin Replacement therapy In Septic Shock (ARISS) study will investigate whether maintenance of serum albumin levels of ≥3.0 g/dL with HAS for 28 days improves survival in patients with septic shock when compared with resuscitation and volume maintenance without albumin [215]. A single-center RCT in an adult surgical population is investigating the effect of goal-directed albumin substitution (aiming at a serum level >3.0 g/dL) and will analyze the frequency of postoperative complications using the Postoperative Morbidity Survey (POMS), which includes an infectious domain [216].

## 8. Conclusions

Hypoalbuminemia is associated with the acquisition and severity of viral, bacterial, and fungal infections and predicts infectious complications in non-infective disease. Systemic inflammation in severe infection alters the function and kinetics of albumin, which in turn can increase the risk of worse clinical outcome. This is confirmed by prospective RCTs at low risk of bias showing that supplementation of albumin with HAS can affect innate and adaptive immune responses with beneficial outcome. In COVID-19, interaction of albumin with bioactive lipid mediators may play a particular role in the occurrence of cytokine storm and organ failure. Hypoalbuminemia alters pharmacokinetics and pharmacodynamics of antimicrobial drugs, necessitating therapeutic drug monitoring and measurement of albumin serum levels, as has been recommended in guidelines for optimizing treatment. Infusion of HAS effectively supplements functioning albumin molecules, which is of proven benefit for infection control in patients with cirrhosis. Prospective RCTs are underway that will investigate the effects of HAS infusion on infectious diseases in patients with hypoalbuminemia. If hypotheses are confirmed, these studies could lead to a change in clinical practice for the management of hypoalbuminemia in patients at risk from infections.

## Figures and Tables

**Figure 1 ijms-22-04496-f001:**
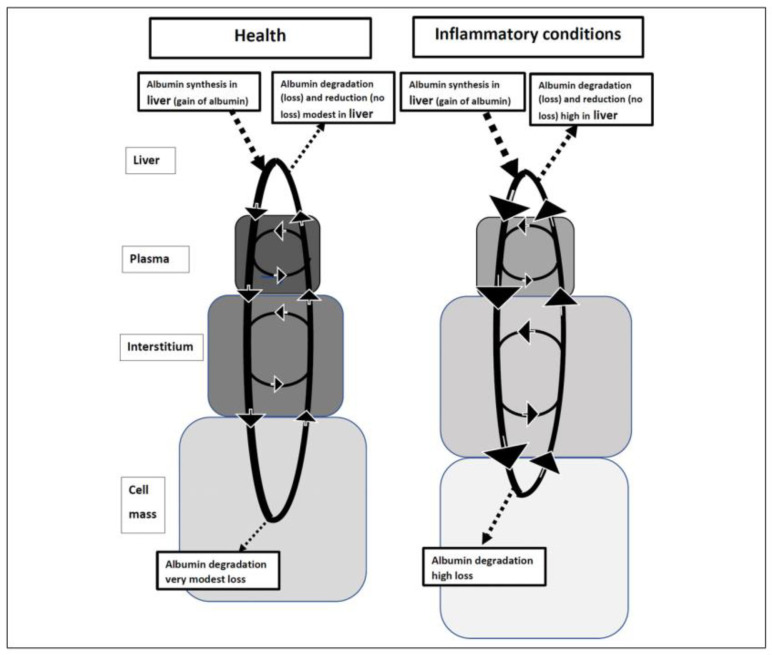
Schematic showing albumin synthesis and degradation in health and inflammatory conditions. Solid arrows represent trans-membrane transport and dashed arrows represent synthesis in the liver. Degradation refers to intracellular proteolysis, Gain to increased total body albumin, Loss to decreased total body albumin, and Reduction to reducing oxidized total body albumin. Reproduced without modification from Soeters et al. 2019 [4], under a Creative Commons Attribution-NonCommercial-NoDerivatives 4.0 International (CC BY-NC-ND 4.0) License (https://creativecommons.org/licenses/by-nc-nd/4.0/ (accessed on 25 April 2021)). Copyright © 2021 The Authors. This reuse has not been endorsed by the licensor. The source reference is “Hypoalbuminemia: Pathogenesis and Clinical Significance” in *JPEN J Parenter Enteral Nutr* and is available at https://aspenjournals.onlinelibrary.wiley.com/doi/full/10.1002/jpen.1451 (accessed on 25 April 2021).

**Figure 2 ijms-22-04496-f002:**
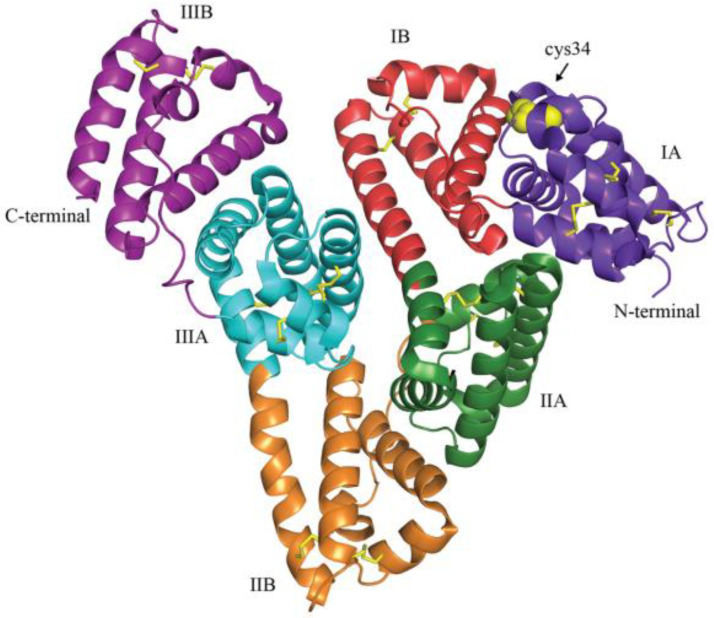
Illustration of the tertiary structure of human serum albumin. The three domains of albumin are shown in purple (IA), red (IB), green (IIA), orange (IIB), blue (IIIA), and violet (IIIB). Yellow sticks depict disulfide bridges, and yellow spheres highlight the available cysteine 34 in domain IA. Reproduced without modification from Larsen et al. 2016 [34], under a Creative Commons Attribution 4.0 International (CC BY 4.0) License (https://creativecommons.org/licenses/by/4.0/ (accessed on 25 April 2021)). No copyright, as the Creative Commons Public Domain Dedication waiver (https://creativecommons.org/publicdomain/zero/1.0/ (accessed on 25 April 2021)) applies. This reuse has not been endorsed by the original authors. The source reference is “Albumin-based drug delivery: harnessing nature to cure disease” in *Mol Cell Ther* and is available at https://molcelltherapies.biomedcentral.com/articles/10.1186/s40591-016-0048-8 (accessed on 25 April 2021).

**Figure 3 ijms-22-04496-f003:**
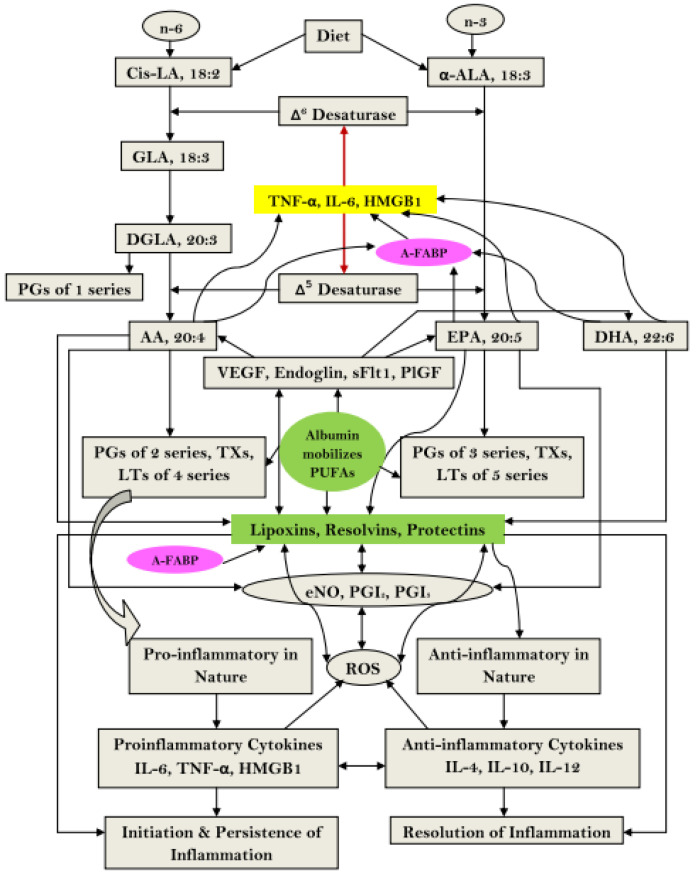
Schematic showing the role albumin in the metabolism of essential fatty acids and their role in inflammation. AA, arachidonic acid; A-FABP, adipose-fatty acid binding protein; ALA, alpha-linolenic acid; DGLA, dihomo-gamma-linolenic acid; DHA, docosahexaenoic acid; eNO, endothelial nitric oxide; EPA, eicosapentaenoic acid; GLA, gamma-linolenic acid; HMGB1, high-mobility group box 1; IL, interleukin; LA, linolenic acid; LT, leukotriene; PG, prostaglandin; PlGF, placental growth factor; PGI, prostacyclin; PUFA, polyunsaturated fatty acid; ROS, reactive oxygen species; sFlt1, Soluble fms-like tyrosine kinase 1; TNFα, tumor necrosis factor alpha; TX, thromboxane; VEGF, vascular endothelial growth factor. Reproduced without modification from Das 2015 [58], under a Creative Commons Attribution 4.0 International (CC BY 4.0) License (https://creativecommons.org/licenses/by/4.0/ (accessed on 25 April 2021)). No copyright, as the Creative Commons Public Domain Dedication waiver (https://creativecommons.org/publicdomain/zero/1.0/ (accessed on 25 April 2021)) applies. This reuse has not been endorsed by the original authors. The source reference is “Albumin infusion for the critically ill–is it beneficial and, if so, why and how?” in Crit Care and is available at https://ccforum.biomedcentral.com/articles/10.1186/s13054-015-0862-4 (accessed on 25 April 2021).

## Data Availability

No new data were created or analyzed in this study. Data sharing is not applicable to this article.

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
