# Peer review of "Hypoalbuminemia as Surrogate and Culprit of Infections"

_ijms, 2021, doi:10.3390/ijms22094496_

Round 1
Reviewer 1 Report
The effects of albumine on infectious disease are predominantly indirect and might reflect the poor conditions the respective patient is in. Subsequently, (critical) ill patients such as those suffering from hypalbunemia, likely habe also immune-mediated defects that are not secondary to hypalbunemia. The authors should adress these points in the discussion. Furthermore, they should indicate whether there is a direct link between albumine and DISTINCT immune responses.
Please also address, whether
hypalbunemia is associated with infections with specific pathogens; and
there are any known mechanisms driving hypalbunemia in C. diff. infections?
Reviewer 2 Report
The paper represents a comprehensive contribution to the role of human serum albumine in human health. It shows how hypoalbuminemia influences a lot of deseases including infections. There are not major and minor imperfections in the manuscript.
Reviewer seems that the manuscript has been preferentialy oriented on medical aspect of albumine and molecular view is emphasized less. This fact is not in coincidence with scope of the IJMS exactly. Reviewer would appreciate explanations of the albumine affecting processes from view of albumine molecule changes to approach the IJMS scope much better.
Author has used a lot of fresh references for detailed issue description doing the contribution modern and necessary especially for physicians.
The manuscript is organized perfectly and I found only one formal mistake on page 4, row 2: physiologic = physiological.
Author Response
see uploaded file
